

# An invitation to the principal series

**Tarek Anous[1⋆] and Jim Skulte[2†]**

**1** Institute for Theoretical Physics and Δ-Institute for Theoretical Physics,
University of Amsterdam, Science Park 904, 1098 XH Amsterdam, The Netherlands
**2** Institute for Theoretical Physics and Center for Extreme Matter and Emergent Phenomena,
Utrecht University, 3508 TD Utrecht, The Netherlands

⋆ t.m.anous@uva.nl, † j.p.skulte@uu.nl

## Abstract

Scalar unitary representations of the isometry group of $d$-dimensional de Sitter space $SO(1,d)$ are labeled by their conformal weights $\Delta$. A salient feature of de Sitter space is that scalar fields with sufficiently large mass compared to the de Sitter scale $1/\ell$ have *complex* conformal weights, and physical modes of these fields fall into the unitary continuous principal series representation of $SO(1,d)$. Our goal is to study these representations in $d=2$, where the relevant group is $SL(2,\mathbb{R})$. We show that the generators of the isometry group of $dS_2$ acting on a massive scalar field reproduce the quantum mechanical model introduced by de Alfaro, Fubini and Furlan (DFF) in the early/late time limit. Motivated by the ambient $dS_2$ construction, we review in detail how the DFF model must be altered in order to accommodate the principal series representation. We point out a difficulty in writing down a classical Lagrangian for this model, whereas the canonical Hamiltonian formulation avoids any problem. We speculate on the meaning of the various de Sitter invariant vacua from the point of view of this toy model and discuss some potential generalizations.

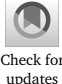

# 1   Introduction

An elegant feature of general relativity is the appearance of certain universal geometries at the edges of parameter space. An intimately familiar instance of this phenomenon is the emergence of an $AdS_2$ throat in the near horizon region of black holes cooled to zero temperature, in *any* spacetime dimension. $AdS_2$ is so pertinent that uncovering its underlying microscopic nature was deemed imperative early on [1–6], with renewed interest more recently [7–14]. A guiding principle in the study of $AdS_2$ is its $SL(2,\mathbb{R})$ group of isometries, which are used to classify the particle states in, and spacetime excitations around, the background geometry.

There exists a similarly universal geometry, which also exhibits an $SL(2,\mathbb{R})$ symmetry, that emerges at the edge of parameter space when the cosmological constant is positive. This spacetime is $dS_2$ and it appears in the near horizon region of black holes in de Sitter space whose Schwarzschild radius is pushed towards the de Sitter horizon, again in any spacetime dimension. These are known as Nariai black holes [15,16], and their holography has also received some interest [17,18] (see [19–24] for recent work relevant to $dS_2$).

Since time immemorial, we have used the fact that the AdS isometry group coincides with that of a conformal field theory in one less dimensions. Exploring the allowed set of particle dynamics in AdS thus amounts to mapping out the space of consistent conformal field theories. The latter has now been codified as "the bootstrap program" [25–29]; and for various important reasons [30,31], the starting point is always a Hamiltonian bounded from below such that the dynamics generated and mediated through interactions is stable. Particle states in this case are labeled by their conformal dimensions $\Delta$ which are taken to be real and positive.

However, in de Sitter space for fields with mass-squared above the de Sitter scale, the conformal dimensions are complex. We will review this fact below. This is not surprising from a group theoretic standpoint; $SL(2,\mathbb{R})$ admits unitary representations with complex conformal weights. These representations are known as the continuous principal series [32,33], an intriguing feature of which is that the $L_0$ spectrum is unbounded above and below, meaning it does not fall within the framework of the conventional bootstrap. Nevertheless the principal series representation makes an appearance in numerous settings of physical interest:

- As one of the allowed boundary modes of $AdS_2$ [34].

- In the operator spectrum of complex SYK [35].

- In the spectrum of particle states in the ergo-region of a rotating black hole, indicating the onset of superradiance [14,36].

- In the celestial decomposition of scattering states [37–39].

- In the decomposition of conformal four-point functions [40].

Given the ubiquity of the principal series, it would be useful to have a toy model to ground ourselves. The simplest example would be an $SL(2,\mathbb{R})$ invariant quantum mechanics, such as the de Alfaro, Fubini, Furlan (DFF) model [41]. Depending on the value of $\Delta$ appearing in the Hamiltonian, this model is known to fill out the discrete and continous series representations of $SL(2,\mathbb{R})$, but not the principal series representation. In fact, the DFF model must be modified in order to accommodate the principal series [42,43], a fact that we will review in detail below. We will also show how the modified DFF model is born out by the dS$_2$ isometries acting on a massive scalar field, at late times.

The structure of our paper is as follows: section 2 starts with a review of the unitary representations of $SL(2,\mathbb{R})$. In section 3 we study the DFF model of [41], suitably altered to accommodate the principal series [42, 43]. In section 4 we describe the geometry of dS$_2$ in global coordinates, including its isometries. In section 5 we study free massive scalar field theory in dS$_2$ with $m^2\ell^2 > 1/4$. We solve for the modes at late times and show that the dS$_2$ isometries acting on these late time modes reduces to the DFF model discussed in the previous section. We end this section by speculating on a possible analog of the family of de Sitter invariant vacua for the DFF model. Section 6 is saved for speculation and future directions.

## 2 Review of unitary irreducible representations of $SL(2,\mathbb{R})$

Our goal in this section is to briefly review the unitary irreducible representations of $SL(2,\mathbb{R})$, first classified in [32,33]. Both dS$_2$ and AdS$_2$ have an isometry algebra given by $SL(2,\mathbb{R})$, but an important distinction is that the states in AdS$_2$ fall under irreducible representations of the universal cover $\widetilde{SL}(2,\mathbb{R})$. Since this is a subtle point, we will use this section to point out some of the key differences that arise between $SL(2,\mathbb{R})$ and its universal cover.

We start with the generators of $SL(2,\mathbb{R})$ which satisfy:

$$[D,H] = iH , \qquad [D,K] = -iK , \qquad [K,H] = 2iD . \tag{2.1}$$

While much of this section is abstract, we have in mind that $H$ is the Hamiltonian, $D$ generates dilatations and $K$ is the special conformal generator. The group elements all commute with the following quadratic Casimir:

$$C_2 \equiv \frac{1}{2}(HK + KH) - D^2 . \tag{2.2}$$

We will label the eigenvalues of $C_2$ as $\Delta(\Delta-1)$ with $\Delta$ the *conformal dimension* that labels the representation. To build such a representation, we follow [44–46] and start with a conformal primary, which is a state annihilated by $K$:[1]

$$K|0\rangle = 0 , \qquad D|0\rangle = i\Delta|0\rangle . \tag{2.3}$$

Starting from the primary, we use the Hamiltonian to 'translate' the state:

$$|t\rangle \equiv e^{-iHt}|0\rangle . \tag{2.4}$$

Using the Baker-Campbell-Hausdorff equation in conjunction with the algebra (2.1) gives:

$$H|t\rangle = i\partial_t|t\rangle , \qquad D|t\rangle = i(t\partial_t + \Delta)|t\rangle , \qquad K|t\rangle = i\left(t^2\partial_t + 2t\Delta\right)|t\rangle . \tag{2.5}$$

---

[1]Usual presentations take $D$ to be anti-Hermitian, with its eigenvalue $\Delta$ real (see e.g. [44]). In our conventions, $D$ is Hermitian, which explains the appearance of the factor of $i$ when acting on a primary.

It will be convenient to work in a basis of energy eigenstates $|E\rangle$. These can be obtained from the $|t\rangle$ states by Fourier transforming:

$$|E\rangle \equiv \int_{-\infty}^{\infty} dt\, e^{iEt} |t\rangle \,. \tag{2.6}$$

We can integrate by parts to reveal that the action of the algebra on this basis is:

$$H|E\rangle = E|E\rangle\,, \qquad D|E\rangle = -i(E\partial_E + \Delta_s)|E\rangle\,, \qquad K|E\rangle = -\left(E\partial_E^2 + 2\Delta_s\partial_E\right)|E\rangle\,, \tag{2.7}$$

where

$$\Delta_s \equiv 1 - \Delta \tag{2.8}$$

is known as the *shadow* conformal dimension. In order to classify the unitary representations, we must define an inner product such that the operators in (2.1) are self-adjoint. For $H$ this implies:

$$0 = \langle E'|H|E\rangle - \langle E'|H|E\rangle = (E' - E)\langle E'|E\rangle\,,$$

where we act with $H$ on the left in the first term, whereas in the second $H$ acts on the right. From this equality we determine

$$\langle E'|E\rangle = f(E)\delta(E - E')\,. \tag{2.9}$$

The same analysis with the dilatation operator $D$ further constrains the inner product as follows:

$$0 = \langle E'|D|E\rangle - \langle E'|D|E\rangle = (i(E'\partial_{E'} + \Delta_s^*) + i(E\partial_E + \Delta_s))\langle E'|E\rangle$$
$$\hookrightarrow (E'\partial_{E'} + E\partial_E)\langle E'|E\rangle = -(\Delta_s + \Delta_s^*)\langle E'|E\rangle\,.$$

From this we conclude

$$f(E) = |E|^{1-\Delta_s-\Delta_s^*}[c_-\Theta(-E) + c_+\Theta(E)]\,, \tag{2.10}$$

where $\Theta(x)$ is the Heaviside step function and $c_\pm$ are arbitrary constants. Finally the same analysis on the special conformal transformation gives:

$$(\Delta_s - \Delta_s^*)(1 - \Delta_s - \Delta_s^*) = 0\,. \tag{2.11}$$

The final condition leaves us with two options: either $\Delta_s \in \mathbb{R}$ or $\Delta_s = \frac{1}{2}(1 - i\nu)$ with $\nu \in \mathbb{R}$, the latter is known as the principal series representation, and is the main focus of this paper. Note that these are the only two choices that result in a *real* Casimir eigenvalue $\Delta(\Delta - 1)$. Also note that in (2.10), we have allowed for the inner product to distinguish between negative energy states and positive energy states. This is motivated by the fact that the original DFF model [41] analyzes a system where $c_- = 0$, meaning the Hilbert space is spanned only by positive energy states.

A generic state can be written as a superposition of energy eigenstates

$$|\psi\rangle = \int dE\, \psi(E)|E\rangle \tag{2.12}$$

and the above analysis gives that the inner product between any two states is:

$$\langle\chi|\psi\rangle = \int dE\, f(E)\chi^*(E)\psi(E)\,, \quad f(E) = [c_-\Theta(-E) + c_+\Theta(E)]\begin{cases} |E|^{2\Delta-1} & \Delta \in \mathbb{R} \\ 1 & \Delta \in \frac{1}{2}(1 + i\nu) \end{cases}. \tag{2.13}$$

The action of the generators (2.7) on the wavefunctions $\psi(E)$ can be infered by integration by parts:

$$H\psi(E) = E\psi(E)\,, \qquad D\psi(E) = i(E\partial_E + \Delta)\psi(E)\,, \qquad K\psi(E) = -(E\partial_E^2 + 2\Delta\partial_E)\psi(E)\,. \quad (2.14)$$

A crucial feature in this construction appears if we want to introduce a 'position' basis, which we will label as $|x\rangle$, along with a resolution of the identity

$$\int dx |x\rangle\langle x| = 1\,. \quad (2.15)$$

In this basis the inner product is then

$$\langle \chi|\psi\rangle = \int\int dx\, dy\, \langle y|x\rangle \chi^*(y)\psi(x) \quad (2.16)$$

and the overlap $\langle y|x\rangle$ is determined by the particular position space representation we choose for the energy eigenstates, along with compatibility with (2.13). The content of this paper will rely on the possibility of interesting position bases for the generators of $SL(2,\mathbb{R})$.

**Ladder operators and normalizability**

The standard way of classifying the unitary $SL(2,\mathbb{R})$ representations starts by defining the following operators

$$L_0 = \frac{1}{2}(H+K)\,, \quad (2.17)$$

$$L_\pm = \frac{1}{2}(H-K) \mp iD\,. \quad (2.18)$$

We now proceed as follows: we take the generator $L_0$ to be compact, meaning it has a discrete spectrum and its eigenvalues are integers. The operators $L_\pm$ raise and lower this eigenvalue by 1. A crucial difference between the case at hand and the universal cover $\widetilde{SL}(2,\mathbb{R})$ is that the universal cover allows the eigenfunctions of $L_0$ to not be single valued, meaning its eigenvalues are not necessarily integers.

The Hilbert space is spanned by states $\psi_n(E)$ that satisfy:

$$L_0\psi_n = -n\psi_n\,, \qquad L_\pm\psi_n = -(n\pm\Delta_s)\psi_{n\pm 1}\,. \quad (2.19)$$

The general solution to these equations is:

$$\psi_n(E) = \int_{-\infty}^{\infty} dt (1+it)^{-n-\Delta_s}(1-it)^{n-\Delta_s} e^{-iEt}\,. \quad (2.20)$$

We must now determine whether these states are normalizable with respect to the inner product (2.13).

**Principal Series:** $\Delta = \frac{1}{2}(1+i\nu)$

The states with $\Delta = \frac{1}{2}(1+i\nu)$ are normalizable for any integer $n$ if $f(E) = 1$ in (2.13). As mentioned before, this representation is known as the principal series and will be our main concern in what follows.

**Complementary series:** $\Delta \in \mathbb{R}$

We now move on to the case of $\Delta \in \mathbb{R}$, where we have to determine the conditions for normalizability of the wavefunctions. Without loss of generality, we restrict to the case $n = 0$, as states with different $n$ can be obtained by repeated application of $L_{\pm}$. The Fourier transform (2.20) can be readily done and gives

$$\psi_0(E) \propto |E|^{\frac{1}{2}-\Delta} K_{\frac{1}{2}-\Delta}(|E|) , \qquad (2.21)$$

where $K_\nu(x)$ is a modified Bessel function. It is easy to check that normalizability with respect to (2.13) (for any choice of positive $c_{\pm}$) requires $0 < \Delta < 1$. Thus states in the complementary series are labeled by $n \in \mathbb{Z}$ and $0 < \Delta < 1$.

**Discrete series:** $\Delta_s \in \mathbb{Z}^+$

The last case pertains to $\Delta_s \in \mathbb{Z}^+$. If this is the case, two more representations can be defined: those in the discrete series. These are often called highest/lowest weight representations. It is straightforward to infer by looking at (2.19) that for $n = \pm \Delta_s$:

$$L_{\pm} \psi_{n = \mp \Delta_s} = 0 . \qquad (2.22)$$

These states are explicitly given by

$$\psi_{n = \pm \Delta_s}(E) \propto |E|^{2\Delta_s - 1} e^{\mp E} \Theta(\pm E) \qquad (2.23)$$

and are normalizable with respect to (2.13) (again with arbitrary choices for $c_{\pm}$) so long as $\Delta_s \in \mathbb{Z}^+$. The highest weight states are obtained by acting with powers of $L_-$ on $\psi_{n = -\Delta_s}$. These states are spanned by $n = -\Delta_s, -\Delta_s - 1 \ldots$. The lowest weight states are obtained similarly by acting with $L_+$ on $\psi_{n = \Delta_s}$ with eigenvalues labeled by the set $n = \Delta_s, \Delta_s + 1, \ldots$. We pause here to mention that the discrete series representations only existing for integer $\Delta_s$ may seem unfamiliar in the context of AdS. This is because the relevant group in AdS$_2$ is actually the universal cover $\widetilde{SL}(2, \mathbb{R})$ and we need not have integer $\Delta_s$ in this case. For dS$_2$ this implies that the masses of e.g. scalars need to be tachyonic and fine tuned in order to expect these representations to appear.

**Summary**

These exhaust the unitary irreducible representations of $SL(2, \mathbb{R})$. We provide a summary of these representations in the following table for convenience:

| Representation | Range of $\Delta$ | Range of $n$ |
|---|---|---|
| Principal series | $\Delta = \frac{1}{2}(1 + i\nu)$ with $\nu \in \mathbb{R}$ | $n \in \mathbb{Z}$ |
| Complementary series | $0 < \Delta < 1$ | $n \in \mathbb{Z}$ |
| Discrete highest weight | $\Delta_s = 1 - \Delta \in \mathbb{Z}^+$ | $n = -\Delta_s, -\Delta_s - 1, -\Delta_s - 2 \ldots$ |
| Discrete lowest weight | $\Delta_s = 1 - \Delta \in \mathbb{Z}^+$ | $n = \Delta_s, \Delta_s + 1, \Delta_s + 2 \ldots$ |

In the next section we will review a particular model that furnishes the principal series representation of this algebra.

## 3 The DFF model and the principal series representation

In this section, we work with a particular representation of the operators (2.1). Precisely, we will consider the following $SL(2,\mathbb{R})$ generators acting on wavefunctions of a single degree of freedom $\theta \in [0, 2\pi)$:

$$H = 2i \cos\left(\frac{\theta}{2}\right)\left[\Delta \sin\left(\frac{\theta}{2}\right) - \cos\left(\frac{\theta}{2}\right)\partial_\theta\right],\tag{3.1}$$

$$K = -2i \sin\left(\frac{\theta}{2}\right)\left[\Delta \cos\left(\frac{\theta}{2}\right) + \sin\left(\frac{\theta}{2}\right)\partial_\theta\right],\tag{3.2}$$

$$D = -i\left[\Delta \cos\theta + \sin\theta\, \partial_\theta\right].\tag{3.3}$$

These operators satisfy the algebra:

$$[D, H] = iH, \qquad [D, K] = -iK, \qquad [K, H] = 2iD.\tag{3.4}$$

We will show that this is actually none other than the de Alfaro, Fubini, Furlan (DFF) model [41] in disguise—suitably altered to accommodate the the principal series representation. This observation was first made in [42, 43]. The reader may recall that the original DFF model describes a particle moving in a repulsive $1/r^2$ potential. We will make contact with this presentation in section 3.1.

The Hilbert space of this model fills out one of the unitary irreducible representations of $SL(2,\mathbb{R})$ labeled by $\Delta$. As we summarized in the previous section, the representation is unitary if the quadratic Casimir

$$C_2 = \frac{1}{2}(HK + KH) - D^2 = \Delta(\Delta - 1)\tag{3.5}$$

is real, and if the operators are self-adjoint with respect to a particular inner product. Let us pick the standard inner product on Hilbert space:

$$(f, g) = \int_0^{2\pi} d\theta\, f^*(\theta) g(\theta),\tag{3.6}$$

then the operators (3.1-3.3) are self adjoint with respect to this inner product if and only if $\Delta = \frac{1}{2}(1 + i\nu)$ is in the principal series.

**Hilbert space labeled by $L_0$ eigenvalues**

Recall that to build the Hilbert space, we construct the compact operator $L_0$ and raising/lowering operators $L_\pm$. For the case at hand, these are :

$$L_0 = \frac{1}{2}(H + K) = -i\partial_\theta, \qquad L_\pm = \frac{1}{2}(H - K) \mp iD = e^{\mp i\theta}(\mp\Delta - i\partial_\theta),\tag{3.7}$$

with $\Delta = \frac{1}{2}(1 + i\nu)$ for the remainder of the paper. The Hilbert space is spanned by the $L_0$ eigenstates $\psi_n(\theta)$ satisfying:

$$L_0\psi_n(\theta) = -n\psi_n(\theta), \qquad L_\pm\psi_n(\theta) = -(n \pm \Delta)\psi_{n\pm1}(\theta),\tag{3.8}$$

with $n \in \mathbb{Z}$. The wavefunctions are easy to compute:

$$\psi_n(\theta) = \frac{1}{\sqrt{2\pi}}e^{-in\theta}\tag{3.9}$$

and are orthonormal with respect to the standard inner product

$$(\psi_k, \psi_n) = \int_0^{2\pi} d\theta\, \psi_k^*(\theta)\psi_n(\theta) = \delta_{kn}\,. \tag{3.10}$$

Note that $n$ can be any integer, so the state space is unbounded above and below. We also have the following completeness relation:

$$\sum_{n=-\infty}^{\infty} \psi_n^*(\theta)\psi_n(\theta') = \delta(\theta - \theta')\,. \tag{3.11}$$

Phrased in a slightly different manner, we can label the set of principal series states by a ket $|n\rangle$ such that

$$\langle\theta|n\rangle = \frac{1}{\sqrt{2\pi}}e^{-in\theta}\,. \tag{3.12}$$

The completeness relation is then the standard one:

$$\langle\theta|\theta'\rangle = \delta(\theta - \theta')\,. \tag{3.13}$$

### $H$ and $K$ eigenstates

Instead of working with the basis of $L_0$ eigenstates, we could instead work with $H$ or $K$ eigenstates. The wavefunctions

$$\chi_E(\theta) = \frac{1}{2\sqrt{\pi}}e^{iE\tan\left(\frac{\theta}{2}\right)}\left[\cos\left(\frac{\theta}{2}\right)\right]^{-2\Delta} \tag{3.14}$$

satisfy

$$(H - E)\chi_E(\theta) = 0\,, \qquad (\chi_{E'}, \chi_E) = \delta(E - E')\,. \tag{3.15}$$

Note that these wavefunctions are singular at $\theta = \pi$, but are nevertheless delta-function normalizable. They are the continuous Fourier modes of the energy-basis on the circle. It is also straightforward to show that:

$$\int_{-\infty}^{\infty} dE\, \chi_E^*(\theta)\chi_E(\theta') = \delta(\theta - \theta')\,. \tag{3.16}$$

(We remind the reader that $\Delta^* = 1 - \Delta$). Similarly, the wavefunctions

$$\rho_\kappa(\theta) = \frac{1}{2\sqrt{\pi}}e^{-i\kappa\cot\left(\frac{\theta}{2}\right)}\left[\sin\left(\frac{\theta}{2}\right)\right]^{-2\Delta} \tag{3.17}$$

satisfy

$$(K - \kappa)\rho_\kappa(\theta) = 0\,, \qquad (\rho_{\kappa'}, \rho_\kappa) = \delta(\kappa - \kappa')\,. \tag{3.18}$$

As well as

$$\int_{-\infty}^{\infty} d\kappa\, \rho_\kappa^*(\theta)\rho_\kappa(\theta') = \delta(\theta - \theta')\,. \tag{3.19}$$

Given the above properties we can define the following transform and its inverse:

$$\psi(\kappa) = \int_0^{2\pi} d\theta\, \rho_\kappa^*(\theta)\psi(\theta)\,, \qquad \psi(\theta) = \int_{-\infty}^{\infty} d\kappa\, \rho_\kappa(\theta)\psi(\kappa)\,. \tag{3.20}$$

### 3.1 Standard presentation of the DFF model

The classic DFF paper [41] describes the following $SL(2, \mathbb{R})$ invariant quantum mechanics, with generators

$$H = \frac{1}{2}\left[-\partial_r^2 + \frac{(4\Delta - 1)(4\Delta - 3)}{4r^2}\right], \tag{3.21}$$

$$K = \frac{r^2}{2}, \tag{3.22}$$

$$D = -\frac{i}{2}\left(r\,\partial_r + \frac{1}{2}\right), \tag{3.23}$$

where $r$ is a radial variable $r > 0$. For $\Delta(\Delta - 1) \geq -1/4$, the above Hamiltonian describes the radial dynamics of a charged particle particle interacting with a magnetic monopole at the origin [47]. On the other hand, as described in [47, 48], for $\Delta = \frac{1}{2}(1 + i\nu)$, the potential

$$V = -\frac{\nu^2 + \frac{1}{4}}{2r^2} \tag{3.24}$$

is attractive, and in this case, the Hamiltonian operator $H$ fails to be self-adjoint with respect to the inner product on the half-line:

$$\int_0^\infty dr\, f^*(r)g(r) \tag{3.25}$$

and thus this model does not seem to accomodate the principal series representation as a Hilbert space. Fortunately or unfortunately, this is precisely the representation we are interested in.

Based on the discussion in the previous section [42, 43] we notice that the tension arises from the fact that (3.22) fixes the eigenvalues of $K$ to be positive definite. However, note that the completeness relation (3.19) required the eigenvalues of $K$ in the principal series to be valued in $\kappa \in (-\infty, \infty)$. In this sense, the coordinate $\kappa$, being the eigenvalue of $K$, is a coordinate on the representation, and the principal series representation is two-sided.

Thus to make contact with the principal series version of this model in these coordinates, we will define $\kappa = \text{sign}(r)r^2/2$ and take $-\infty < r < \infty$. To see how this works, let us work in the $\kappa$ basis and define a modified transform:

$$\hat{\psi}(\kappa) = |2\kappa|^{\frac{3}{4} - \Delta} \int_0^{2\pi} d\theta\, \rho_\kappa^*(\theta)\psi(\theta) \tag{3.26}$$

along with a new inner product

$$(\hat{\psi}, \hat{\phi})' = \int_{-\infty}^\infty \frac{d\kappa}{|2\kappa|^{1/2}} \hat{\psi}^*(\kappa)\hat{\phi}(\kappa). \tag{3.27}$$

This norm is selected such that the overlap is preserved

$$(\hat{\psi}(\kappa), \hat{\phi}(\kappa))' = (\psi(\theta), \phi(\theta)). \tag{3.28}$$

Acting on the function space $\hat{\psi}(\kappa)$ the operators take the form:

$$H = \frac{1}{2}\left[-2\kappa\partial_\kappa^2 - \partial_\kappa + \frac{(4\Delta - 1)(4\Delta - 3)}{8\kappa}\right], \tag{3.29}$$

$$K = \kappa, \tag{3.30}$$

$$D = -i\left(\kappa\,\partial_\kappa + \frac{1}{4}\right). \tag{3.31}$$

We can now make contact with the DFF model by taking $\kappa = \text{sign}(r)r^2/2$. We find that acting on functions of $r$, the operators take the form

$$H = \frac{\text{sign}(r)}{2}\left[-\partial_r^2 + \frac{(4\Delta - 1)(4\Delta - 3)}{4r^2}\right], \tag{3.32}$$

$$K = \text{sign}(r)\frac{r^2}{2}, \tag{3.33}$$

$$D = -\frac{i}{2}\left(r\,\partial_r + \frac{1}{2}\right), \tag{3.34}$$

thus, in order for the standard DFF model to represent the Hilbert space of the principal series, we must extend $r$ to negative values, and for $r < 0$ *the Hamiltonian flips sign*. This is reminiscent of a horizon in general relativity, although it is difficult to speculate too much at present.

Note that the inner product induced from (3.27) implies wavefunctions will be normalized with respect to the standard $L_2$ norm

$$\int_{-\infty}^{\infty} dr f^*(r)g(r). \tag{3.35}$$

Computing wavefunctions in the $r$ basis is now a simple matter of transforming them from the $\theta$ basis. And we emphasize, despite the bizarre behavior of the Hamiltonian across $r = 0$, that the system is completely unitary. We provide expressions for the eigenstates of the DFF model in the $r$ coordinate in appendix A.

We note here that the transform from the $\theta$ variable to the $r$ variable given through (3.20) is reminiscent of the non-local map from dS to AdS described in [49].

**Obstruction to supersymmetrization**

As a paranthetical, we mention an obstruction to supersymmetrizing this model. In a follow-up to the original DFF paper [41], [50] gave a supersymmetrization of the DFF model. We may thus ask if this construction works for the principal series DFF model described above. This would be unexpected given the general obstructions regarding unitary de Sitter superalgebras [51,52]. Nevertheless, conformal field theories are able to get around this obstruction [53], and even [52] identified the dS$_2$ superalgebra as a special case since its bosonic subgroup is the same as AdS$_2$'s.

However, following the steps in [50], one notices that role of the spin quantum number is played by $\Delta$ and the action of the supercharge $Q$ shifts $\Delta$ by $1/2$, taking us out of the principal series. This obstruction was noted in [54] and it remains unclear to us whether there is a way around it. It may be possible to supersymmetrize a model furnishing the complementary series, at least in the range $0 < \Delta < 1/2$.

## 3.2 Classical and path integral descriptions

What classical dynamical system gives rise to the DFF model with quantum operators (3.1-3.3)? For this, one could imagine sticking with the $r$-variable, for which there is a 'standard' Lagrangian. But as we emphasized, the coordinate singularity at the origin means these variables only cover phase space patchwise.

The most natural choice is to simply use the $\theta$ variable and consider the following functions on phase space $(\theta, p)$ which can be obtained by considering the symplectic structures on the

group manifold $SO(1,2)$ [42]

$$H = 2\cos\left(\frac{\theta}{2}\right)\left[-\nu\sin\left(\frac{\theta}{2}\right) + \cos\left(\frac{\theta}{2}\right)p\right] , \tag{3.36}$$

$$K = 2\sin\left(\frac{\theta}{2}\right)\left[\nu\cos\left(\frac{\theta}{2}\right) + \sin\left(\frac{\theta}{2}\right)p\right] , \tag{3.37}$$

$$D = \nu\cos\theta + \sin\theta\, p . \tag{3.38}$$

In the above expressions, $\nu$ is the classical analog of the conformal dimension, which in the quantum case is given by $\Delta = \frac{1}{2}(1 + i\nu)$, and $p$ is the canonical momentum conjugate to $\theta$. To see this, we define the Poisson bracket

$$\{f,g\} = \partial_\theta f\, \partial_p g - \partial_p f\, \partial_\theta g , \tag{3.39}$$

such that $p$ and $\theta$ form a canonical pair $\{\theta, p\} = 1$. With this, the above set of functions, although linear in $p$ and therefore unbounded, define a classical dynamical system with an $SL(2,\mathbb{R})$ symmetry apparent from its Poisson bracket algebra:

$$\{D,H\} = H , \qquad \{D,K\} = -K , \qquad \{K,H\} = 2D . \tag{3.40}$$

Note also that the combination

$$HK - D^2 = -\nu^2 \tag{3.41}$$

is a constant and therefore conserved with respect to dynamical evolution generated by any possible linear combination of $H$, $D$ and $K$.

Now we are at a crossroads when considering dynamical evolution. None of the operators (3.36-3.38) is a natural choice of time evolution operator, since they are all unbounded above and below. This is in stark contrast to the highest weight representation. For the quantum problem in the highest weight representation, the natural choice is the linear combination defining $L_0$ whose spectrum is bounded and discrete, although any combination of dynamics is equally valid and related by a time reparametrization [41, 47].

**Dynamics generated by $L_0$**

For simplicity we consider the classical dynamics generated by $L_0$:

$$L_0 = \frac{1}{2}(H + K) = p \tag{3.42}$$

$$L_\pm = \frac{1}{2}(H - K) \mp iD = e^{\mp i\theta}(p \mp i\nu) , \tag{3.43}$$

which satisfy

$$\{L_+, L_-\} = -2iL_0 , \qquad \{L_\pm, L_0\} = \mp iL_\pm . \tag{3.44}$$

The classical solutions to $\frac{d\cdot}{dt} - \{\cdot, L_0\} = 0$ are:

$$L_0(t) = p(t) = l_0 , \qquad L_\pm(t) = l_\pm e^{\mp it} , \qquad \theta(t) = \theta_0 + t , \tag{3.45}$$

with $l_\pm = e^{\mp i\theta_0}(l_0 \mp i\nu)$.

This Hamiltonian system is linear in the momentum $p$, so it is difficult to pass to the the Lagrangian picture. To make this clear, let us pass to the quantum path integral for this dynamical system:

$$\langle\theta_f|e^{-iL_0 T}|\theta_i\rangle = \int_{\theta(0)=\theta_i}^{\theta(T)=\theta_f} Dp D\theta \exp\left[i\int_0^T dt\, p\left(\dot\theta - 1\right)\right] = \int_{\theta(0)=\theta_i}^{\theta(T)=\theta_f} D\theta\, \delta\left(\dot\theta - 1\right) , \tag{3.46}$$

where the final equality comes from integrating out $p$. In the Lagrangian presentation, the $\theta$ path integral localizes!

We can compute the path integral by any means, such as spectral decomposition:

$$\langle \theta_f | e^{-iL_0 T} | \theta_i \rangle = \delta(\theta_f - \theta_i - T) \, . \tag{3.47}$$

The purpose of this little exercise are two-fold. First we showed that the quantum dynamics are criminally uninteresting. More importantly is that there is no simple, local Lagrangian giving rise to them.

## Dynamics generated by $\frac{1}{2}(H - K)$

There is another natural choice of dynamics, hinted at by the dS$_2$ construction which we will give in the next section. The analog of the generator of static patch time, which is a boost in global coordinates, is given by the combination $K^2 \equiv \frac{1}{2}(H - K)$. We will organize the operators as

$$L_0 = \frac{1}{2}(H + K) = p \, , \tag{3.48}$$

$$K^2 = \frac{1}{2}(H - K) = -v \sin \theta + \cos \theta \, p \, , \tag{3.49}$$

$$D = v \cos \theta + \sin \theta \, p \, , \tag{3.50}$$

which satisfy the canonical Poisson bracket algebra:

$$\{L_0, K^2\} = D \, , \qquad \{D, K^2\} = L_0 \, , \qquad \{L_0, D\} = -K^2 \, . \tag{3.51}$$

The classical solutions to $\frac{d\cdot}{d\tau} - \{\cdot, K^2\} = 0$ are:

$$L_0(\tau) = p(\tau) = l_0 \cosh \tau + d \sinh \tau \, , \qquad D(\tau) = d \cosh \tau + l_0 \sinh \tau \, . \tag{3.52}$$

Again to obtain a Lagrangian, we look at the quantum path integral:

$$\langle \theta_f | e^{-iK^2 T} | \theta_i \rangle = \int_{\theta(0)=\theta_i}^{\theta(T)=\theta_f} Dp D\theta \exp\left[ i \int_0^T d\tau \left\{ p(\dot{\theta} - \cos \theta) + v \sin \theta \right\} \right]$$

$$= \int_{\theta(0)=\theta_i}^{\theta(T)=\theta_f} D\theta \, \delta\left( \dot{\theta} - \cos \theta \right) e^{i \int_0^T d\tau \, v \sin \theta} \, . \tag{3.53}$$

This can be calculated explicitly, giving

$$\langle \theta_f | e^{-iK^2 T} | \theta_i \rangle = (\cosh T + \sin \theta_i \sinh T)^{\Delta - 1} \, \delta\left( \theta_f - 2\tan^{-1}\left[ \frac{\sinh \frac{T}{2} + \cosh \frac{T}{2} \tan \frac{\theta_i}{2}}{\cosh \frac{T}{2} + \sinh \frac{T}{2} \tan \frac{\theta_i}{2}} \right] \right) \, . \tag{3.54}$$

The late time limit of the above equation is

$$\langle \theta_f | e^{-iK^2 T} | \theta_i \rangle \sim e^{-(1-\Delta)T} \delta(\theta_f - \pi/2), \tag{3.55}$$

which seems to suggest that, according to this dynamics, localized wavepackets tend towards the 'horizon' at $\theta = \pi/2$.

Note in this example the Lagrangian again localizes, but in the presence of a "line operator." Perhaps this suggests that this description is emergent out of something more fundamental [55–57]. Indeed, a path integral of this type was considered in [34], where they found it necessary to replace smooth paths with jagged ones. Perhaps something similar is in order here.

# 4 The spacetime dS$_2$

In this section we review the basic geometric features of dS$_2$, including its ambient space construction and isometries. Once the basics of the geometry are laid out, we will show how the DFF model of the previous section arises when we consider a massive scalar field theory in dS$_2$.

## 4.1 The geometry

The metric in global coordinates is given by

$$ds^2 = -d\tau^2 + \ell^2 \cosh^2\left(\frac{\tau}{\ell}\right) d\theta^2 \,, \tag{4.1}$$

where $\tau$ is the global time coordinate which ranges between $\tau \in (-\infty, \infty)$ and $\theta \sim \theta + 2\pi$ is a coordinate parametrizing a spatial $S^1$. The parameter $\ell$ is the de Sitter length. This metric on dS$_2$ can be constructed by considering a hyperboloid in an ambient 3-dimensional Minkowski space satisfying

$$-\left(X^0\right)^2 + \left(X^1\right)^2 + \left(X^2\right)^2 = \ell^2 \,. \tag{4.2}$$

The metric (4.1) is then induced from the 3d Minkowski metric on the solution to (4.2)

$$X^0 = \ell \sinh\left(\frac{\tau}{\ell}\right) \,, \ X^1 = \ell \cos\theta \, \cosh\left(\frac{\tau}{\ell}\right) \,, \ X^2 = \ell \sin\theta \, \cosh\left(\frac{\tau}{\ell}\right) \,. \tag{4.3}$$

For the sake of completeness, we include the inverse relations:

$$\tau = \ell \, \text{arcsinh}\left(\frac{X^0}{\ell}\right) \,, \qquad \theta = \arctan\left(\frac{X^2}{X^1}\right) \,. \tag{4.4}$$

## 4.2 The isometries

The isometries of dS$_2$ are inherited from the isometries of the hyperboloid (4.2). These include the rotation:

$$J^3 = -i\left(X^1\partial_{X^2} - X^2\partial_{X^1}\right) \,, \tag{4.5}$$

and boosts

$$K^1 = -i\left(X^0\partial_{X^1} + X^1\partial_{X^0}\right) \,, \quad K^2 = -i\left(X^0\partial_{X^2} + X^2\partial_{X^0}\right) \,, \tag{4.6}$$

of the ambient Minkowski space.

Written in terms of the global coordinates, these are:

$$J^3 = -i\,\partial_\theta \,, \tag{4.7}$$

$$K^1 = -i\left(\ell \cos\theta \, \partial_\tau - \sin\theta \tanh\left(\frac{\tau}{\ell}\right)\partial_\theta\right) \,, \tag{4.8}$$

$$K^2 = -i\left(\ell \sin\theta \, \partial_\tau + \cos\theta \tanh\left(\frac{\tau}{\ell}\right)\partial_\theta\right) \,. \tag{4.9}$$

These can be combined into

$$L_0 = J^3 = -i\partial_\theta \,, \tag{4.10}$$

$$L_\pm = K^2 \pm iK^1 = e^{\mp i\theta}\left(-i\tanh\left(\frac{\tau}{\ell}\right)\partial_\theta \pm \ell\partial_\tau\right) \,, \tag{4.11}$$

which satisfy the $SL(2, \mathbb{R})$ algebra

$$[L_+, L_-] = 2L_0 \,, \qquad [L_\pm, L_0] = \pm L_\pm \,. \tag{4.12}$$

This algebra admits a quadratic Casimir that commutes with all the elements:

$$C_2 \equiv L_0^2 - \frac{1}{2}(L_- L_+ + L_+ L_-) \tag{4.13}$$

and unitary irreducible representations are labeled by real eigenvalues of the quadratic Casimir

$$C_2 = \Delta(\Delta - 1) . \tag{4.14}$$

It is clear from the above construction that $L_0$ is a compact generator of the dS$_2$ isometries (it generates a rotation) and therefore should have a discrete spectrum. On the other hand, $K^2$, is a boost,[2] so its spectrum is necessarily continuous.

We can also consider the other canonical basis for the algebra $SL(2,\mathbb{R})$ which we write now:

$$H \equiv L_0 + \frac{1}{2}(L_+ + L_-) , \qquad K \equiv L_0 - \frac{1}{2}(L_+ + L_-) , \qquad D \equiv \frac{i}{2}(L_+ - L_-) . \tag{4.15}$$

We also note here that the boost generator $K^2$ is given by:

$$K^2 = \frac{1}{2}(L_+ + L_-) = \frac{1}{2}(H - K) , \tag{4.16}$$

as we anticipated at the end of the previous section.

# 5 Scalar field theory in dS$_2$

We will now attempt to make contact with the DFF model by studying a simple scalar field theory in two-dimensional de Sitter space. We first proceed in steps, beginning with a review of the classical scalar modes, then continuing on to quantization. We will see that the dS$_2$ isometries acting on late time solutions of the scalar field equations precisely reproduce the operators of the DFF quantum mechanical model.

## 5.1 Classical solutions

We are now ready to use all of this to study a simple quantum field theory in dS$_2$. That of a free scalar with action:

$$S = -\frac{1}{2} \int d^2 x \sqrt{-g} \left[ g^{\mu\nu} \partial_\mu \phi \partial_\nu \phi + m^2 \phi^2 \right] . \tag{5.1}$$

The equations of motion obtained from varying the above action with respect to $\phi$ are

$$\frac{1}{\sqrt{-g}} \partial_\mu \sqrt{-g} g^{\mu\nu} \partial_\nu \phi = m^2 \phi . \tag{5.2}$$

It is not difficult to verify that the scalar Laplacian is none other than the quadratic Casimir operator of the dS$_2$ isometries (4.13)

$$\frac{1}{\sqrt{-g}} \partial_\mu \sqrt{-g} g^{\mu\nu} \partial_\nu \equiv -\frac{1}{\ell^2} C_2 , \tag{5.3}$$

thus, we expect based on the representation theory of $SL(2,\mathbb{R})$ to identify

$$\Delta(\Delta - 1) = -m^2 \ell^2 \tag{5.4}$$

---

[2]The boost $K^2$ can be identified with the generator of static patch time in certain coordinate choices.

or

$$\Delta_\pm = \frac{1}{2}\left(1 \pm \sqrt{1 - 4m^2\ell^2}\right). \tag{5.5}$$

As in AdS, $\Delta_\pm$ label the two possible falloffs (now in time) of the scalar field $\phi$ of mass $m$. Interestingly, if $m^2\ell^2 > 1/4$, the falloffs $\Delta_\pm$ become complex:

$$\Delta_\pm = \frac{1}{2}(1 \pm i\nu), \tag{5.6}$$

with $\nu \in \mathbb{R}$. As should be clear by now, the complex weights are no cause for concern. The Casimir is, after all, real, and the states belong precisely to the continuous principal series representations of $SL(2,\mathbb{R})$, which are unitary. In this case, since $\Delta_\pm$ are complex conjugates, they actually represent the same state in the Hilbert space. We will label $\Delta \equiv \Delta_+$ throughout.

## 5.2 Behavior near $\tau \to \infty$

Let us consider the Klein-Gordon equation for the scalar field in the late-time limit. Despite the explicit time dependence in the metric, we will assume first, and later justify, the following ansatz:

$$\phi(\tau,\theta) \underset{\tau\to\infty}{\approx} f\left(\frac{\tau}{\ell}\right)\psi(\theta). \tag{5.7}$$

Acting on this ansatz, the scalar wave equation (5.2) behaves as

$$\psi(\theta)\left[-\Delta(\Delta-1)f\left(\frac{\tau}{\ell}\right) + \tanh\left(\frac{\tau}{\ell}\right)f'\left(\frac{\tau}{\ell}\right) + f''\left(\frac{\tau}{\ell}\right)\right] = f\left(\frac{\tau}{\ell}\right)\text{sech}^2\left(\frac{\tau}{\ell}\right)\psi''(\theta), \tag{5.8}$$

where $'$ denotes a derivative with respect to the argument. Thus in the $\tau \to \infty$ limit, the right hand side tends to zero and the Klein-Gordon equation is solved, to leading order, by

$$\phi(\tau,\theta) \underset{\tau\to\infty}{\approx} \psi(\theta)\left(c_1 e^{-\Delta\tau/\ell} + c_2 e^{-(1-\Delta)\tau/\ell}\right). \tag{5.9}$$

Recall that since $\Delta = (1 + i\nu)/2$, it satisfies $1 - \Delta = \Delta^*$. We will now judiciously choose our falloff conditions such that, $c_1 = \sqrt{\frac{2}{\nu}}$ and $c_2 = 0$. The Klein-Gordon inner-product between two modes then reduces to:

$$(\phi_1,\phi_2) = -i\int_\Sigma d\Sigma^\mu\left(\phi_1\partial_\mu\phi_2^* - \phi_2^*\partial_\mu\phi_1\right) = \int_0^{2\pi} d\theta\,\psi_1(\theta)\psi_2^*(\theta), \tag{5.10}$$

which is presicely the $L_2$ inner product on wavefunctions of a single compact degree of freedom. We will take the suggestion given to us by the geometry very seriously.

We end this section by writing down the action of the dS$_2$ isometries (4.10-4.11) on the modes that falloff as $e^{-\Delta\tau/\ell}$ in the $\tau \to \infty$ limit. These are:

$$L_0 = -i\partial_\theta\,, \qquad L_\pm = e^{\mp i\theta}(\mp\Delta - i\partial_\theta)\,. \tag{5.11}$$

We immediately recognize these as the generators of the DFF quantum mechanics (3.7). We will quantize the scalar field in the next section. We note here that the inner product (5.10) implies the following Hermitian conjugates for the operators

$$L_0^\dagger = L_0\,, \qquad L_\pm^\dagger = L_\mp\,. \tag{5.12}$$

### 5.3 Quantization

We now move on to a discussion of the quantized scalar field in $dS_2$ given the above realizations, following [49, 58, 59]. We can decompose our scalar field in modes

$$\phi(\tau,\theta) = \sum_{n=-\infty}^{\infty} a_n\, \phi_n^{\text{out}}(\tau,\theta) + a_n^\dagger \phi_n^{*\text{out}}(\tau,\theta), \qquad (5.13)$$

with

$$\phi_n^{\text{out}}(\tau,\theta) \equiv f_n^{\text{out}}(\tau)\psi_n(\theta), \qquad (5.14)$$

and $\psi_n(\theta)$ as in (3.9). The solutions $f_n^{\text{out}}$ are chosen such that they decay as

$$\lim_{\tau\to\infty} f_n^{\text{out}}(\tau) \approx \sqrt{\frac{2}{\nu}} e^{-\Delta\tau/\ell}\ . \qquad (5.15)$$

Explicitly, we find

$$f_n^{\text{out}}(\tau) = 2^{|n|}\sqrt{\frac{2}{\nu}} e^{-(|n|+\Delta)\frac{\tau}{\ell}} \cosh^{|n|}\left(\frac{\tau}{\ell}\right) {}_2F_1\left(\frac{1}{2}+|n|, |n|+\Delta, \frac{1}{2}+\Delta, -e^{-2\frac{\tau}{\ell}}\right)\ . \qquad (5.16)$$

The modes satisfy

$$(\phi_n^{\text{out}}, \phi_m^{\text{out}}) = \delta_{nm}\ , \qquad (\phi_n^{\text{out}}, \phi_m^{*\text{out}}) = 0\ , \qquad (5.17)$$

with respect to the Klein-Gordon norm and it is easy to verify that for generators $(L_0, L_\pm)$ given in (4.10)-(4.11):

$$L_0\phi_n = -n\phi_n\ , \qquad L_\pm\phi_n = -(n\pm\Delta)\phi_{n\pm 1}\ . \qquad (5.18)$$

It is now clear that the single-particle Hilbert space of this scalar field is in one-to-one correspondence with the Hilbert space of the DFF model in the principal series representation.

To canonically quantize this theory, as usual, we promote the coefficients $a_n$ and $a_n^\dagger$ to creation annihilation operators satisfying:

$$[a_n, a_m^\dagger] = \delta_{nm}\ , \qquad [a_n, a_m] = [a_n^\dagger, a_m^\dagger] = 0\ , \qquad (5.19)$$

and denote a vacuum state $|0\rangle_{\text{out}}$ such that

$$a_n|0\rangle_{\text{out}} = 0 \qquad \forall n\ . \qquad (5.20)$$

Now if we compute the two-point correlation function in the state $|0\rangle_{\text{out}}$, we find:

$$\lim_{\tau,\tau'\to\infty} {}_{\text{out}}\langle 0|\phi(\tau,\theta)\phi(\tau',\theta')|0\rangle_{\text{out}} = \frac{2}{\nu} e^{-\Delta\frac{\tau-\tau'}{\ell}-\frac{\tau'}{\ell}}\delta(\theta-\theta')\ . \qquad (5.21)$$

In [58] it was suggested this meant that the interactions at $\mathcal{I}^+$ are ultralocal in the state $|0\rangle_{\text{out}}$, but we may similarly interpret this in a more mundane manner, in light of (3.11) it seems that the equal time Green's function is computing the completeness relation of the coordinate basis Hilbert space .

### 5.4 Euclidean modes

The vacuum $|0\rangle_{\text{out}}$ annihilated by the modes $\phi_n^{\text{out}}$ is but one of a family of de Sitter invariant vacua. These are known as the Motolla-Allen, or $\alpha$, vacua [60,61]. Among these, the Euclidean or Bunch-Davies vacuum plays a prominent role for its particular entanglement structure [58,

62]. The Euclidean vacuum is annihilated by modes that are analytic on the lower half-sphere in the Euclidean continuation of the global coordinates (4.1). These modes are [58]:

$$\phi_n^E(\tau, \theta) = f_n^E(\tau)\psi_n(\theta),$$ (5.22)

with

$$f_n^E(\tau) = \frac{i(-2)^{-|n|}}{\Gamma(1+|n|)}\sqrt{\frac{(1-e^{\pi\nu})\,\pi\Gamma(|n|+1-\Delta)\Gamma(|n|+\Delta)}{-2i(1-2\Delta)\Gamma\left(\frac{1}{2}-\Delta\right)\Gamma\left(-\frac{1}{2}+\Delta\right)}}$$

$$e^{(|n|+1-\Delta)\frac{\tau}{\ell}}\cosh^{|n|}\left(\frac{\tau}{\ell}\right){}_2F_1\left(\frac{1}{2}+|n|,|n|+\Delta,1+2|n|,1+e^{2\frac{\tau}{\ell}}\right).$$ (5.23)

By expanding at late times, it is not so difficult to verify that

$$\phi_n^{\mathrm{out}} = \frac{e^{i\gamma_n}}{\sqrt{1-e^{\pi\nu}}}\left(\phi_n^E - e^{\frac{\pi\nu}{2}}\phi_{-n}^{*E}\right),$$ (5.24)

where the phase $e^{i\gamma_n}$ is

$$e^{2i\gamma_n} = \frac{\Gamma\left(\frac{1}{2}-\Delta\right)\Gamma(\Delta+|n|)}{\Gamma\left(-\frac{1}{2}+\Delta\right)\Gamma(1-\Delta+|n|)}.$$ (5.25)

This can be inverted to give

$$\phi_n^E = \frac{1}{\sqrt{1-e^{\pi\nu}}}\left(e^{-i\gamma_n}\phi_n^{\mathrm{out}} - e^{\frac{\pi\nu}{2}}e^{i\gamma_n}\phi_{-n}^{*\mathrm{out}}\right).$$ (5.26)

We can now choose to expand our scalar field in these modes

$$\phi(\tau, \theta) = \sum_{n=-\infty}^{\infty} b_n\phi_n^E(\tau, \theta) + b_n^\dagger\phi_n^{*E}(\tau, \theta),$$ (5.27)

where the coefficients $b_n, b_n^\dagger$ get promoted to operators upon quantization, with:

$$[b_n, b_m^\dagger] = \delta_{nm}, \qquad [b_n, b_m] = [b_n^\dagger, b_m^\dagger] = 0.$$ (5.28)

The state annihilated by $b_n$ is the Euclidean vacuum:

$$b_n|E\rangle = 0 \qquad \forall n.$$ (5.29)

Now from (5.24), we can read off

$$a_n \equiv \frac{e^{i\gamma_n}}{\sqrt{1-e^{\pi\nu}}}\left(b_n - e^{\frac{\pi\nu}{2}}b_{-n}^\dagger\right),$$ (5.30)

which gives

$$\lim_{\tau\to\infty}\langle E|\phi(\tau, \theta)\phi(\tau, \theta')|E\rangle = \frac{2}{\nu}e^{-\frac{\tau}{\ell}}\frac{1+e^{\pi\nu}}{1-e^{\pi\nu}}\delta(\theta-\theta')$$

$$-\frac{e^{\frac{\pi\nu}{2}}}{1-e^{\pi\nu}}\sum_{m=-\infty}^{\infty}\frac{e^{-2\Delta\frac{\tau}{\ell}}}{\pi\nu}\frac{\Gamma\left(\frac{1}{2}-\Delta\right)\Gamma(\Delta+|m|)}{\Gamma\left(-\frac{1}{2}+\Delta\right)\Gamma(1-\Delta+|m|)}e^{-im(\theta-\theta')} + \mathrm{cc}.$$ (5.31)

The above equation is the late time limit of the Euclidean two-point function $G_E$, meaning we can obtain the sum using standard techniques [63]. Dropping the $\delta$-function singularity gives:

$$\lim_{\tau\to\infty}G_E = e^{-2\Delta\frac{\tau}{\ell}}\frac{4^{-(1-\Delta)}\Gamma(\Delta)\Gamma(1-2\Delta)}{\pi\Gamma(1-\Delta)}\left|\sin\left(\frac{\theta-\theta'}{2}\right)\right|^{-2\Delta} + (\Delta\to 1-\Delta).$$ (5.32)

This expression for the two-point function is real, as expected for a real scalar field, but we could alternatively construct the following complex field, as in Appendix B of [36]:

$$s(\tau, \theta) = \sum_{n=-\infty}^{\infty} a_n \phi_n^{\text{out}}(\tau, \theta),$$ (5.33)

which, by construction, has a single complex fall-off in its late time two-point function:

$$\lim_{\tau \to \infty} \langle E | s(\tau, \theta) s(\tau, \theta') | E \rangle = C \, e^{-2\Delta \frac{\tau}{\ell}} \left| \sin\left(\frac{\theta - \theta'}{2}\right) \right|^{-2\Delta}.$$ (5.34)

### 5.5 $\alpha$-vacua in the DFF model?

In the hopes of building up a dictionary, we can ask how to compute the Euclidean two-point function (5.32) or the the correlator of complex operators $s(\tau, \theta)$ in the state $|E\rangle$ from the point of view of the DFF model— i.e. with the global time dependence stripped off. This may give us some insight into what the Motolla-Allen vacua [60, 61] correspond to in the putative holographic dual. And while (5.34) may look like a completeness relation from the point of view of $SL(2, \mathbb{R})$ it is good to remember that such a completeness relation only holds for for certain choices of representations and choices of position space basis. For example, a particular choice for the complementary series in [33] gives a completeness relation that resembles (5.34), but as we described in section 2, these have $0 < \Delta < 1$.

To this end, we can try and guess the answer by using the bulk as our guide. Consider a slightly modified basis of wavefunctions for the DFF model which differs only by a phase

$$\tilde{\psi}_n(\theta) \equiv e^{i\gamma_n} \psi_n(\theta),$$ (5.35)

with $\psi_n$ given in (3.9) and the phase specified in (5.25). The natural expectation is for these phases to change very little in terms of the physics. The Bunch-Davies two-point function (5.32), however, suggests we consider the following real combination:

$$\frac{1}{2} \sum_{n=-\infty}^{\infty} \tilde{\psi}_n(\theta) \tilde{\psi}_{-n}(\theta') + \tilde{\psi}_n^*(\theta) \tilde{\psi}_{-n}^*(\theta') = \frac{\pi^{1/2} \Gamma(\Delta)}{\Gamma\left(-\frac{1}{2} + \Delta\right)} \left| \sin\left(\frac{\theta - \theta'}{2}\right) \right|^{-2\Delta} + \text{cc.}$$ (5.36)

This clearly gives an $SL(2, \mathbb{R})$ invariant function of the points $(\theta, \theta')$, but note that this expression is quite different from the completeness relation we found before in (3.11):

$$\langle \theta | \theta' \rangle = \delta(\theta - \theta').$$ (5.37)

To see what modifications are needed to obtain this result, let us consider arranging our states in a doublet $|n\rangle$ defined such that such that :

$$\langle \theta | n \rangle \equiv \frac{1}{\sqrt{2}} \left[ \tilde{\psi}_n(\theta) |0\rangle + \tilde{\psi}_n(\theta)^* |1\rangle \right],$$ (5.38)

where we have tensored our principal series Hilbert space with a two-dimensional 'qubit' Hilbert space. Let us also suggest a modification of the adjoint:

$$\begin{aligned}
(n | \theta \rangle &\equiv \frac{1}{\sqrt{2}} \left[ \tilde{\psi}_n(-\theta) \langle 0 | + \tilde{\psi}_n^*(-\theta) \langle 1 | \right], \\
&= \frac{1}{\sqrt{2}} \left[ \tilde{\psi}_{-n}(\theta) \langle 0 | + \tilde{\psi}_{-n}^*(\theta) \langle 1 | \right].
\end{aligned}$$ (5.39)

Then (5.36) simply becomes[3]

$$\widetilde{\langle\theta|\theta'\rangle} \equiv \text{Tr}_{0,1} \sum_n \langle\theta|n)(n|\theta'\rangle = \frac{\pi^{1/2}\Gamma(\Delta)}{\Gamma\left(-\frac{1}{2}+\Delta\right)} \left|\sin\left(\frac{\theta-\theta'}{2}\right)\right|^{-2\Delta} + \text{cc.} \tag{5.40}$$

This way of presenting things suggests that (5.36) could be a completeness relation in a modified sense.

Studying equation (4.9) of [58] suggests the following potential generalization of the above completeness relation corresponding to the $\alpha$-vacua:

$$G_\alpha(\theta,\theta') = \sum_{n=-\infty}^{\infty} \tilde{\psi}_n(\theta)\tilde{\psi}_n^*(\theta') + \frac{1}{2}\left(1-e^{\alpha+\frac{\pi\nu}{2}}\right)\left(1-e^{\alpha^*-\frac{\pi\nu}{2}}\right)\tilde{\psi}_n(\theta)\tilde{\psi}_{-n}(\theta')$$

$$+ \frac{1}{2}\left(1-e^{\alpha-\frac{\pi\nu}{2}}\right)\left(1-e^{\alpha^*+\frac{\pi\nu}{2}}\right)\tilde{\psi}_n^*(\theta)\tilde{\psi}_{-n}^*(\theta') \tag{5.41}$$

and we recover (3.11) for the choice $\alpha = \pi\nu/2$. In this section we are suggesting that these choices amount perhaps to a selection of an inner product on Hilbert space, in a similar spirit to the CPT inner product suggested in [64]. It would be interesting to understand if there exists a principle that fixes this choice corresponding to $\alpha$. In [58] it was inferred that these label a family of theories related by a marginal deformation. If so, perhaps such a deformation can be written down for the DFF model (3.1)-(3.3); but we leave this speculation for future work.

# 6 Future Directions

In this paper we have studied a simple quantum mechanical model whose Hilbert space captures the single particle states of a free massive scalar field in $\text{dS}_2$. In this section we list some related open questions that would be worth visiting in the future.

**Multiparticle generalizations:** The DFF model admits a multiparticle generalization known as the Calogero model [65] whose operators also fulfill the $SL(2,\mathbb{R})$ algebra [66,67]

$$H = \left[-\frac{1}{2}\sum_i \partial_{x_i}^2 + \sum_{i<j} \frac{\lambda^2}{(x_i-x_j)^2}\right], \tag{6.1}$$

$$K = \sum_i \frac{x_i^2}{2}, \tag{6.2}$$

$$D = -\frac{i}{4}\sum_i \left(x_i\,\partial_{x_i} + \partial_{x_i}x_i\right). \tag{6.3}$$

As far as we are aware, a multipartical generalization of the principal series version of this model, akin to (3.1)-(3.3) has not yet been formulated. More interestingly, there exists a relationship between Calogero type models and the eigenvalue dynamics of matrix models related to gauge theory [68,69]. We have attempted to write down a generalization in this spirit, but have so far only succeeded in writing down the trivial multiparticle generalization of

---

[3]We want to reaffirm that this is not a standard inner product, as for two arbitrary states $|\xi\rangle$ and $|\chi\rangle$ this implies

$$\widetilde{\langle\xi|\chi\rangle} = \frac{1}{2}\int d\theta\, [\xi(-\theta)\chi(\theta) + \xi^*(-\theta)\chi^*(\theta)].$$

(3.1)-(3.3)—that is a sum over single-body generators. We hope to write down the interacting version of the multiparticle principal series Calogero model in the future, and work out its connection to matrix models if such a connection exists.

**2d field theory dual to dS$_3$:**    The global isometries of dS$_3$ consist of two copies of the $SL(2, \mathbb{R})$ algebra. It has been speculated that the asymptotic symmetry group gets enhanced to two copies of the Virasoro algebra [70]. The single particle Hilbert space of a heavy scalar field in dS$_3$ again falls into the principal series representation and it would be interesting to explore the possibility of constructing two-dimensional conformal field theories whose operator content fills out these representations. Some implications of these ideas have been explored in [71,72] and more recently in [73]. We want to note that there exist unitary irreducible principal series representations of the Virasoro algebra, but these require the central charge $c = 0$ [74]. The appearance of two copies of $SL(2, \mathbb{R})$ in dS$_3$ is intricately linked with the fact that the $S^2$ at $\tau \to \infty$ admits a complex structure via the stereographic map. However, it must be said that the individual $L_0$ and $\bar{L}_0$ generators are not compact on their own—only the diagonal element $L_0 + \bar{L}_0$ is. This perhaps suggests, yet again, that we need to think carefully about how to build quantum field theories dual to de Sitter, as these will not be obtained from AdS by a simple analytic continuation.

**Principal series bootstrap:**    This point was emphasized in [36] but nevertheless is worth repeating. While the Euclidean conformal group in $d$ dimensions $SO(1, d + 1)$ admits continuous principal series representations, the choice is usually made to discard these representations from appearing in the crossing equations [75–77] due to their unbounded spectrum. Given these representations' relevance to massive quantum field theory in a fixed de Sitter background, perhaps it would be fruitful to understand how the boostrap equations must be modified to include them. This would then be relevant to the cosmological bootstrap program [78–80].

**Inversion formula:**    Standard conformal field theory is built out of states that furnish the discrete highest weight representation of the conformal algebra. However, it is the conformal blocks in the principal series that form a basis of functions [40]. That is, we can write any four point function in standard Euclidean CFT as:

$$G(z) = 1 + \int_{\frac{1}{2} - i\infty}^{\frac{1}{2} + i\infty} \frac{d\Delta}{2\pi i} c(J, \Delta) F_{J, \Delta}(z) \tag{6.4}$$

and $c(J, \Delta)$ must be an analytic function with poles on the highest weight states. For interacting fields in de Sitter, the internal and external states may themselves live in the principal series, meaning the poles of $c(J, \Delta)$ will generically be shifted. We would like to explore what other ingredients are necessary to ensure a sensible four-point function.

# Acknowledgments

We thank Dionysios Anninos, Austin Joyce, Jorrit Kruthoff, Raghu Mahajan, Greg Mathys, Gui Pimentel, and John Stout for insightful discussions and Dionysios Anninos, Austin Joyce, Raghu Mahajan and John Stout for a careful reading of the draft. TA is supported by the Delta ITP consortium, a program of the Netherlands Organisation for Scientific Research (NWO) that is funded by the Dutch Ministry of Education, Culture and Science (OCW). This research was supported in part by the National Science Foundation under Grant No. NSF PHY-1748958.

## A   Principal series DFF wavefunctions in $r$-space

We collect here expressions for the normalized principal-series wavefunctions for the DFF model using the coordinate $r$. These are obtained by transforming either (3.9) or (3.14) using the kappa-transform (3.26).

### $L_0$ eigenstates

We start with the $L_0$ eigenstates:

$$\psi_n(\theta) = \frac{e^{-in\theta}}{\sqrt{2\pi}} \, ,$$

the $\kappa$ transform of which is given by

$$\hat{\psi}(\kappa) = \frac{2^{\frac{1}{4}-\Delta}e^{-in\pi}}{\pi}|\kappa|^{\frac{3}{4}-\Delta}\int_{-\infty}^{\infty} du\,(1-iu)^{n-\Delta}(1+iu)^{-n-\Delta}e^{-i\kappa u}, \tag{A.1}$$

where we changed coordinates $\theta \to \pi + 2\tan^{-1}u$. This Fourier transform can be found e.g. on page 119 of volume one of [81] and gives:

$$\hat{\psi}_n(r) = \frac{2^{\frac{3}{2}-2\Delta}e^{-in\pi}}{\sqrt{|r|}}\begin{cases} -\frac{W_{-n,\frac{1}{2}-\Delta}(r^2)}{\Gamma(-n+\Delta)} & r > 0 \\ \frac{W_{n,\frac{1}{2}-\Delta}(r^2)}{\Gamma(n+\Delta)} & r < 0 \end{cases}, \tag{A.2}$$

where $W_{n,m}(x)$ are the Whittaker functions. To verify that these are properly normalized, it suffices to use the following identity [82]:

$$\int_0^{\infty} dx\,\frac{W_{k,\mu}(x)W_{\lambda,-\mu}(x)}{x}$$
$$= \frac{\pi}{(k-\lambda)\sin(2\pi\mu)}\left[\frac{1}{\Gamma\left(\frac{1}{2}-k+\mu\right)\Gamma\left(\frac{1}{2}-\lambda-\mu\right)} - \frac{1}{\Gamma\left(\frac{1}{2}-k-\mu\right)\Gamma\left(\frac{1}{2}-\lambda+\mu\right)}\right]. \tag{A.3}$$

### $H$ eigenstates

The $\kappa$-transform of the energy eigenstates can be done straightforwardly in `Mathematica`. We provide the wavefunctions here:

$$\hat{\chi}_E(r) = 2^{\frac{1}{2}-\Delta}e^{-i\pi\Delta}|E|^{\frac{1}{2}-\Delta}\sqrt{|r|}\begin{cases} -J_{-i\nu}\left(\sqrt{2|E|r^2}\right) & E > 0, r > 0 \\ \frac{2\sin(2\pi\Delta)}{\pi}K_{i\nu}\left(\sqrt{2|E|r^2}\right) & E > 0, r < 0 \\ -\frac{2\sin(2\pi\Delta)}{\pi}K_{i\nu}\left(\sqrt{2|E|r^2}\right) & E < 0, r > 0 \\ J_{i\nu}\left(\sqrt{2|E|r^2}\right) & E < 0, r < 0 \end{cases}. \tag{A.4}$$

From these expressions, it is clear that the Hamiltonian flips sign across the origin. At fixed energy, the wavefunctions are oscillatory on one side and decaying on the other. These expressions are slightly different than those found in [43].

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
