# Peer review of "An invitation to the principal series"

_SciPost Physics, doi:SciPost Phys. 9, 028 (2020)_

## Round 2 · Referee Report · Anonymous (Referee 2) · 2020-8-19

Strengths

Provide a novel and synergetic link between different research areas.

Weaknesses

It relies very heavily on some observations obtained in cited papers.

Report

The paper “An invitation to the principal series” (Anous and Skulte) concerns the role of the conformal symmetry in de Sitter space. First, the authors recall some basic facts concerning the unitary representations of the isometry group of dS_2 space, with the special emphasis put on the principal series. Next, following the reference [43] they recall that the principal series can be also described as the (modified) DFF model [41]. On the other hand, the authors consider the late time limit of the scalar field in dS_2 space. Since the isometries acting on late time solutions of the scalar field carry (for some parameters) the principal series representation (cf. [59]) the authors infer that these solutions can be related to the modified DFF model. The last part of the paper includes some considerations concerning various de Sitter invariant vacua in the above context.
I have to admit that the paper contains more questions than answers and relies very heavily on some known observations. However, the paper relates various aspects of the relevant symmetry in consequence it can stimulate further investigations. Moreover, the topics considered are interesting and still very extensively studied (due to their potential applications, e.g, in various dualities, black holes physics and inflationary models). In view of this I think that such “an invitation” can be useful and fruitful, thus I can recommend it for publication in SciPost Physics.

---

## Round 2 · Referee Report · Anonymous (Referee 1) · 2020-8-19

Strengths

Clear presentation and technical exposition.

Weaknesses

At times, the authors could do slightly better highlighting which results / insights are new given the large amount of reviewed content.

Report

Meets all general acceptance criterion of SciPost and has a clear set of interesting followup directions, as well as potential to help link various existing subfields where principal series representations appear. A lot of the value added with respect to the last point comes from the clarity of their review section on SL(2,R) representations. Indeed, as the title suggests, the majority of this paper reads like a cohesive review. The focus of this paper is on a modification to the DFF model for which principal series representations appear. The main new result is showing that this modified principal series DFF model is realized by a massive scalar field in dS2.

Requested changes

N/A

---

## Editorial Decision

published